# Beyond the Prompt: Deploying Medical Foundation Models on Diverse Chest X-ray Populations

**Louisa Fay**[1,2,3] 🆔                    LFAY@STANFORD.EDU
**Jean-Benoit Delbrouck**[1]               JBDEL@STANFORD.EDU
**Thomas Küstner**[2]          THOMAS.KUESTNER@MED.UNI-TUEBINGEN.DE
**Bin Yang**[3]                  BIN.YANG@ISS.UNI-STUTTGART.DE
**Noel C. F. Codella**[4]            NCODELLA@MICROSOFT.COM
**Matthew P. Lungren**[4]            MLUNGREN@MICROSOFT.COM
**Curtis P. Langlotz**[1]              LANGLOTZ@STANFORD.EDU
**Sergios Gatidis**[1]                SGATIDIS@STANFORD.EDU

[1] *Department of Radiology, School of Medicine, Stanford University, CA, USA*

[2] *Medical Image and Data Analysis, University Hospital of Tübingen, Germany*

[3] *Institute of Signal Processing and System Theory, University of Stuttgart, Germany*

[4] *Microsoft Health and Life Sciences, Redmond, WA, USA*

**Editors:** Accepted for publication at MIDL 2025

## Abstract

Foundation models (FMs) have shown impressive performance in medical image analysis tasks, but their deployment in real-world clinical settings, especially across diverse patient populations such as adult and pediatric cases, remains challenging. Key open questions include optimal prompting techniques and strategies for model adaptation or fine-tuning for clinical use. In this study, we evaluated different approaches for deploying FMs in clinical scenarios for diverse patient populations. We use the lightweight, embedding-based vision-language FM *MedImageInsight* to predict pneumonia from chest X-rays, a condition common in both adult and pediatric patients. We observed a large variation in model predictive performance depending on the chosen prompt design, highlighting the importance of text prompt design for successful zero-shot (ZS) application. On in-domain datasets, we found performance differences of up to 46% in Matthews correlation coefficient (MCC) and 56% in true positive rates across different text prompts. By introducing text and vision embedding ensembles, we achieved substantial ZS improvements, outperforming training-based methods (fine-tuning, linear probe) in low-data scenarios by up to 43% for adults and 35% for pediatric populations (MCC). This ensembling strategy also promotes resource-efficient, equitable clinical use by supporting diverse demographic subgroups, achieving MCC improvements of 6% by sex, 17% by age, and 10% by race compared to linear probe.

**Keywords:** multimodal foundation model, bias, zero-shot, pneumonia, ensembles

## 1. Introduction

Foundation Models (FMs) that have been trained on extensive web-based datasets have demonstrated great promise and remarkable generalizability across a variety of tasks in different domains, including natural language processing, computer vision, and text and image generation (Brown et al., 2020; Radford et al., 2021). Similarly, their medical counterparts, trained on domain-specific datasets such as PubMed, electronic health records, and medical

imaging, have shown significant potential to advance healthcare applications (Zhang et al., 2022; Singhal et al., 2023). However, their reliable implementation in clinical settings without further adjustments remains challenging due to the severe consequences of incorrect diagnoses or treatment plans (Huang et al., 2023). While an increasing number of FMs are developed using medical data, their clinical application often experiences performance drops on out-of-distribution data, such as in new patient populations (e.g., transitioning from adult to pediatric cases) (Chen et al., 2024; Zhang et al., 2023; Huang et al., 2024).

Moreover, since many FMs are trained to derive predictions from vision-language similarities, their effective training-free, zero-shot (ZS) application depends not only on the input image but also on the given text prompt. Determining an optimal text prompt to achieve the best ZS performance in various environments, particularly in new distributions, still poses a major challenge.

A common strategy for applying vision-language FMs in clinical settings relies on adapting and fine-tuning image encoders (Chambon et al., 2022; Hu et al., 2021). However, this approach requires additional diverse and labeled data, which is expensive and difficult to acquire. Furthermore, most FMs are based on large transformer models, which require significant computational resources to fine-tune. As many healthcare facilities lack the necessary infrastructure, these approaches are unsuitable for integration into clinical workflows.

Our study aims to address these challenges by identifying effective strategies for the successful clinical application of FMs, focusing on the state-of-the-art, open-source, lightweight, embedding-based vision-language FM, *MedImageInsight* (Codella et al., 2024) which showed superior performance and suitability across multiple tasks and domains. Since *MedImageInsight* was predominantly trained on adult data, this study examines the prediction of pneumonia from chest X-rays in adult (in-domain) and pediatric (out-of-domain) cases using training-free ZS and training-based approaches. Key contributions of our work include:

- **ZS ensemble for training-free FM deployment:** Enhancing ZS prediction by introducing text and vision ensembles for medical tasks.
- **Analysis of FM adaptation trade-offs:** Comparative analysis of ZS, LoRA fine-tuning, and lightweight adapters (linear probe, k-NN), showing that fine-tuning requires sufficient data to be effective, while linear probing can introduce biases.
- **Multi-site evaluation** using MIMIC-CXR (in-domain, adults, part of training data), CheXpert (external data, adults) and VinDr-PCXR (out-of-domain, pediatric) datasets.
- **Bias assessment** of ZS and training-based methods across sex, age, and race groups.

Related works and limitations are provided in Appendix A, B. Code is publicly available[1].

## 2. Methods

We evaluated the open-source FM *MedImageInsight* for pneumonia detection in three different domains by assessing its ZS capability using nine different prompt types and enhanced prompt and vision ensembles (Figure 1) as well as by exploring training-based methods.

*MedImageInsight* (Codella et al., 2024) comprises three parts: an image encoder (360M parameters), a text encoder (250M parameters), and an optional text decoder (70M parameters). We excluded the text decoder, resulting in a lightweight FM with 610M parameters.

---

1. https://github.com/loufay/Beyond-the-prompt

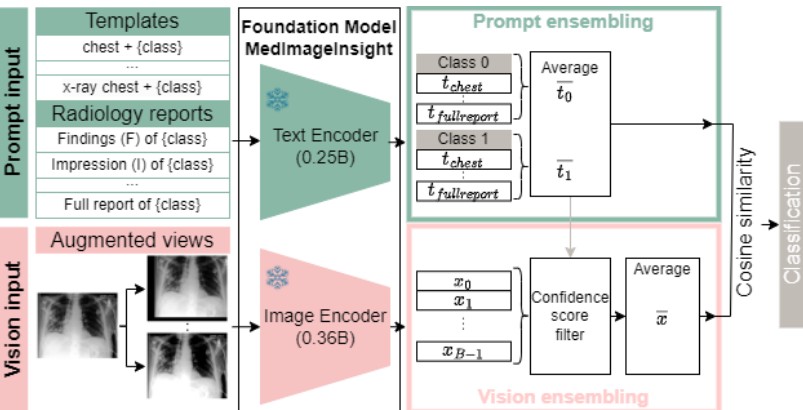

Figure 1: Overview of our Beyond-the-prompt pipeline using a zero-shot ensemble method. (top) Text embeddings are averaged over prompt templates and radiology reports and (bottom) compared to image ensembles generated by augmented views. A confidence-score filter is applied to select the most reliable image embeddings. Cosine similarity between text and image ensembles defines predicted classes.

The model was trained on image-text pairs from 14 modalities, including adult chest X-rays, along with radiology reports from MIMIC-CXR database (Johnson et al., 2019), and image-label pairs from NIH-CXR-LT (Holste et al., 2022) and Mass General Brigham database.

## 2.1. Zero-shot Evaluation

Embedding-based contrastive vision-language FMs compute the cosine similarity between an image embedding and multiple text embeddings to perform training-free ZS classification.

### 2.1.1. Prompt input

**Templates as prompt.**  A common approach of constructing a prompt in ZS classification involves a combination of a *text-prompt template* plus a placeholder, {*class*}, that represents the class names, (here: *No Finding* or *Pneumonia*). The image is assigned to the class of the closest text prompt in the embedding space. We evaluated the FM's performance in predicting pneumonia using the following text prompts: 1) {*class*}, 2) *chest*+{class}, 3) *X-ray*+{class}, 4) *chest X-ray*+{class}, 5) *X-ray chest anteroposterior*+{class}.

**Radiology reports as prompt.**  Since *MedImageInsight* is trained on radiology reports, we additionally evaluate our model by generating text embeddings using textual information extracted from the following parts of the radiology reports: 6) findings section, 7) impression section, 8) both, findings and impression sections, or 9) full radiology reports. We randomly sample ten reports from each class in the training set and compute the distance between the text embeddings generated from these reports and an image embedding from the test set. The predicted class is determined based on the majority of the five closest text embeddings.

**Template and radiology report ensembles as prompt.**  As introduced by (Radford et al., 2021), generating averaged text embeddings can enhance ZS results and reduce

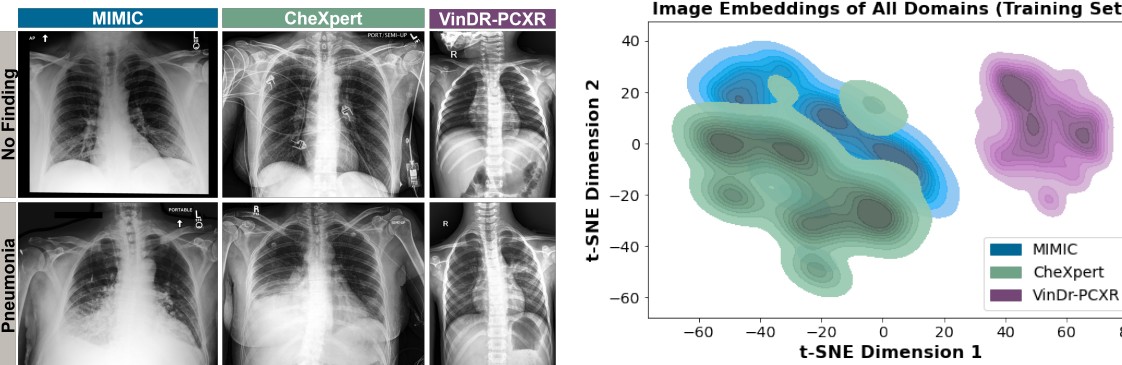

Figure 2: (left) Chest X-ray examples of 'No Finding' and 'Pneumonia' cases across adult (MIMIC, CheXpert) and pediatric (VinDr-PCXR) domains (right) Comparison of X-ray embeddings of *MedImageInsight* across the three domains using t-SNE. Embeddings of the test set and separated by classes are attached in Appendix C.

computational complexity, as a general text ensemble is created once and reused during inference. An averaged text embedding $\overline{t_c} = \frac{1}{P} \sum_{p=1}^{P} t_{cp}$ is computed for each class $c$ using all $P$ prompt embeddings $t$. Most medical FMs are trained on template prompts and radiology reports. Hence, merely averaging the template prompts may not be sufficient. Therefore, we propose an extension by generating: 10) averaged template-based embeddings using the templates (1-5), 11) report-based embeddings using ten reports per class of each report type (6-9), and 12) embeddings that incorporate both template- and report-embeddings (1-9).

### 2.1.2. VISION INPUT

**Original X-ray as vision input.** The most common approach to generate an input embedding is to use the original image, which, in our case, represents a chest X-ray.

**Vision ensemble as vision input.** By augmenting an input image $B$ times and creating $B$ image embeddings, a single, representative embedding can be generated by averaging these $B$ embeddings. This embedding is compared to a given text embedding (Shu et al., 2022; Döbler et al., 2024). This method aims to enhance the robustness and diversity of the image embeddings, potentially improving the alignment with text representations. We chose $B = 64$ (Shu et al., 2022) and applied a random selection of the following augmentation techniques: random rotation within a range of $\pm 10°$, random affine transformations with translation up to 10% of the image dimensions, color jittering (brightness=0.2, contrast=0.2), Gaussian blurring (kernel size = 5), and automatic contrast enhancement.

**Confidence score filtered (CF) vision ensemble as vision input.** In addition to averaging all augmented views, we implement a CF method as in (Shu et al., 2022). This approach identifies the $N$ most confident augmented images using entropy-based confidence filtering by determining the 10% of samples with the lowest entropy.

## 2.2. Training-based adaption

To compare the training-free ZS methods, we investigate training-based adaptation strategies that leverage the image encoder by adding lightweight adapters as well as performing full fine-tuning of the image encoder. Unless stated otherwise, we run each experiment five times with randomly sampled training data and used cross entropy loss and AdamW (Loshchilov, 2017) optimizer with a learning rate of $3 \times 10^{-4}$. If validation performance did not improve for five epochs, the learning rate was reduced by factor of 0.1.

**Lightweight adapter training.** In adapter training, the image encoder remains frozen while the image embeddings are further processed using subsequent lightweight adapter heads. We applied Linear Probing ($R^{1024 \times 2}$) and $k$-nearest neighbor (kNN) ($k = 5$).

**Fine-tuning using Low-Rank Adaptation (LoRA).** The parameters of the image encoder are adapted using LoRA (Hu et al., 2021), which modifies the parameters with a low intrinsic rank updates (rank $r = 8$).

Hyperparameters $k$ and $r$ were empirically selected (Appendix D.1, Table 3 and 4).

**Baseline Models.** We used four baseline models, each from a different model type. (1) DenseNet-121 (Huang et al., 2017), a simple CNN with 7M parameters, 98% smaller than the image encoder of *MedImageInsight*, but showed competitive results in various medical applications (Singh et al., 2024). (2) CheXagent (Chen et al., 2024), an instruction-tuned FM (8B parameters) trained exclusively on chest X-rays to generate free text. The training of CheXagent included all applied three datasets. (3) *RAD-DINO* (Pérez-García et al., 2024) is an image encoder (86.6M) that generates embeddings of size $R^{768 \times 2}$. To enable classification, we trained a linear head (Linear Probing). This model was trained only on medical data including MIMIC and CheXpert datasets. (4) BiomedCLIP (Zhang et al., 2023) (0.09B) is a contrastive vision-language model trained on PubMed data. Its architecture supports the direct application of our zero-shot (ZS) methods.

## 2.3. Datasets

We used three publicly available chest X-ray datasets to evaluate pneumonia prediction in three different environments. Figure 2 (left) shows representative examples for each dataset. To ensure consistency and fairness during training under varying amounts of training data, we balanced all training datasets using 744 samples. We also balanced all test datasets.

- **MIMIC-CXR (in-domain, adults)** (Johnson et al., 2019) was part of the FM training. Our test set included 8,186 X-rays with labels and radiology reports.
- **CheXpert (external validation, adults)** (Irvin et al., 2019) was not part of the training set for *MedImageInsight*, but comprises adult subjects, similar to MIMIC. Our balanced test set contains 2,508 samples. We also generated balanced test sets for the demographic subgroups: sex (male/female - 1151 samples per group), age (young:$< 62$ years/old: $> 62$ years - 1128 samples per group), and race (White/Asian/Black - 171 samples per group) to assess biased prediction differences.
- **VinDr-PCXR (new domain, pediatrics)** (Pham et al., 2022) represents a new domain, as it exclusively contains pediatric cases, which were not part of the training of *MedImageInsight*. The balanced test set contains 178 samples.

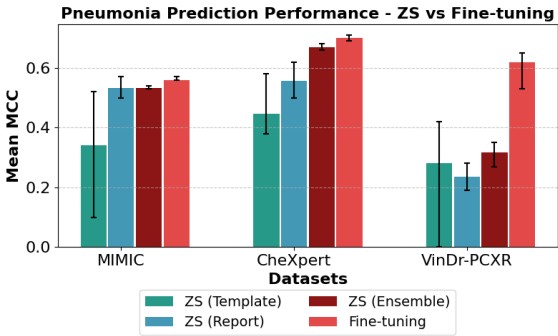 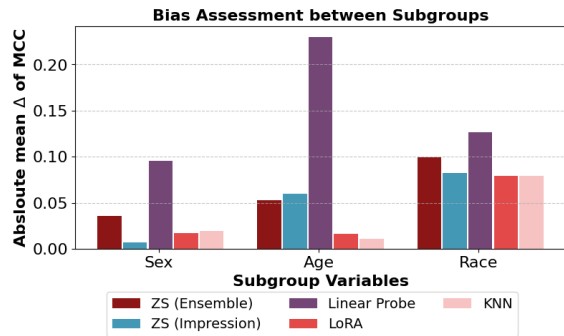

Figure 3: (left) Mean MCC for ZS experiments across different prompt types: (1-5) Template, (6-9) Report, and most effective Ensemble using (12) Template and Report (MIMIC, CheXpert); (10) Templates (VinDr-PCXR) compared to fine-tuning. (right) Bias Assessment: Absolute mean MCC difference across subgroups: sex (Male, Female), age (≤62,>62 years), race (Asian, Black, White).

## 3. Results and Discussion

### 3.1. Zero-shot Evaluation

As highlighted in Figure 3 (left), ZS performance of *MedImageInsight* highly depends on the specific text prompt to which a given X-ray is compared. A detailed evaluation of all prompt types and metrics is depicted in Table 1. Similar ZS behavior was detected for the baseline FM *BiomedCLIP* (Appendix D.1, Table 2).

**Template as prompt.** Among our five types of prompt templates, we obtained the highest accuracy (Acc) on all adult and pediatric datasets by using the prompt template (2) *chest+*{class}. Most other templates resulted in either TPR or TNR below 50%.

**Radiology reports as prompt.** Comparing the image embeddings to parts of the radiology reports revealed notable performance boosts for the adult datasets. MIMIC performed best using (5) *Findings* as prompt (Acc = 78.0%, MCC = 0.57). CheXpert performed slightly better using (6) *Impression* (Acc = 80.5%, MCC = 0.62). While MIMIC and CheXpert achieved comparable results using either (5) *Findings* or (6) *Impression*, operating on full reports led to performance drops of up to 7% in Acc. In contrast, when comparing pediatric cases, VinDr-PCXR, to any part of MIMIC reports, TPR were consistently < 20%.

**Prompt Ensembles.** Using prompt ensembles further improved ZS performance. For adult datasets, using prompt ensembles of (11) *Reports* or (12) *Templates + Reports* was most effective. Although, for MIMIC, using the simple (6) *Findings* prompt yielded slightly higher accuracy and MCC, using prompt ensembles enhanced the TPR by up to 5.4% while maintaining a strong TNR. On VinDr-PCXR, significant improvements were observed using (10) *Template* ensembles, resulting in a TPR increase of 7.3% compared to the best valid TPR of (2) *chest+*{class}.

Table 1: ZS performance for different text and vision prompts across adult (MIMIC, CheXpert), and pediatric (VinDr-PCXR) datasets. (Acc, TNR, TPR in [%].)

| | | Prompt | Vision Ens.[+] | MIMIC Acc | TNR | TPR | MCC | CheXpert Acc | TNR | TPR | MCC | VinDr-PCXR Acc | TNR | TPR | MCC |
|---|---|---|---|---|---|---|---|---|---|---|---|---|---|---|---|
| Template | 1 | | | 59.9 | 98.5 | 21.4 | 0.31 | 62.9 | 99.3 | 26.6 | 0.38 | 60.1 | 80.9 | 39.3 | 0.22 |
| | 2 | *chest* | | 74.8 | 88.9 | 60.7 | 0.52 | 76.4 | 97.2 | 55.6 | 0.58 | 69.1 | 85.4 | 52.8 | 0.40 |
| | 3 | *X-ray* | - | 67.8 | 96.5 | 39.1 | 0.43 | 65.5 | 99.1 | 31.8 | 0.42 | 68.0 | 94.4 | 41.6 | 0.42 |
| | 4 | *x. c.** | | 61.3 | 98.4 | 24.1 | 0.34 | 66.2 | 99.0 | 33.5 | 0.43 | 61.8 | 100 | 23.6 | 0.37 |
| | 5 | *x. c. a.** | | 51.2 | 3.0 | 99.7 | 0.1 | 70.3 | 84.7 | 55.8 | 0.42 | 50.0 | 0 | 100 | 0 |
| Report | 6 | Findings (F) | | 78.0 | 83.8 | 72.2 | 0.57 | 80.1 | 87.9 | 72.4 | 0.61 | 58.0 | 98.7 | 17.2 | 0.27 |
| | 7 | Impression (I) | - | 77.7 | 83.7 | 71.8 | 0.56 | 80.5 | 88.0 | 72.9 | 0.62 | 58.1 | 98.9 | 17.3 | 0.28 |
| | 8 | F+I | | 74.6 | 84.7 | 64.5 | 0.50 | 73.6 | 90.3 | 56.9 | 0.50 | 53.3 | 100 | 7.0 | 0.19 |
| | 9 | Full Report | | 74.7 | 84.8 | 64.5 | 0.50 | 73.5 | 90.3 | 56.6 | 0.50 | 53.9 | 100 | 8.0 | 0.2 |
| Prompt Ensemble | 10 | Templates | | 64.2 | 97.5 | 30.8 | 0.38 | 73.7 | 98.3 | 48.5 | 0.54 | 63.5 | 66.9 | 60.1 | 0.27 |
| | 11 | Reports | - | 74.4 | 71.2 | 77.6 | 0.49 | 82.4 | 85.4 | 79.5 | 0.65 | 55.9 | 25.8 | 86.0 | 0.15 |
| | 12 | Both | | 76.3 | 80.4 | 72.2 | 0.53 | 82.8 | 89.0 | 76.6 | 0.66 | 56.2 | 28.1 | 84.3 | 0.15 |
| | 10 | Templates | All | 64.3 | 97.6 | 31.0 | 0.38 | 70.1 | 99.2 | 40.1 | 0.49 | 66.9 | 77.5 | 56.2 | 0.35 |
| | 11 | Reports | All | 75.0 | 73.2 | 76.7 | 0.5 | 82.9 | 83.8 | 82.0 | 0.66 | 53.9 | 18.0 | 90.0 | 0.11 |
| | 12 | Both | All | 76.5 | 81.8 | 71.3 | 0.53 | 83.7 | 89.0 | 78.3 | 0.68 | 54.8 | 21.9 | 87.6 | 0.12 |
| | 10 | Templates | CF[++] | 65.5 | 97.3 | 33.6 | 0.40 | 71.5 | 99.0 | 44.1 | 0.51 | 66.3 | 74.2 | 58.4 | 0.33 |
| | 11 | Report | CF[++] | 75.1 | 74.3 | 76.0 | 0.50 | 82.7 | 83.9 | 81.6 | 0.65 | 55.6 | 18.0 | 93.2 | 0.17 |
| | 12 | Both | CF[++] | 76.8 | 82.3 | 71.1 | 0.54 | 83.4 | 88.8 | 78.0 | 0.67 | 54.5 | 20.2 | 88.8 | 0.12 |

*x.=X-ray, c.=chest, a.=anteroposterior; [+]Ens.= Ensemble; [++]CF=confidence-score filter

**Prompt and Vision Ensembles.** By additionally generating vision ensembles with augmented views, slight improvements in accuracy were achieved for MIMIC and CheXpert. Specifically, for the new adult domain, CheXpert, TPR was further improved by up to 2.5%. For the pediatric dataset, TPR slightly dropped when using vision ensembles.

**Discussion.** We found that using prompt ensembles is highly valuable and improves performance when applying FMs in ZS settings. Overall, we achieved best performance on CheXpert, followed by MIMIC, and the pediatric dataset VinDr-PCXR. Our results showed that if a given X-ray image belongs to a distribution similar to training (i.e. MIMIC and CheXpert; Figure 2), prompt ensembles that include radiology reports enhance ZS performance. In this case, using vision ensembles further improved performance as more variability was added, better reflecting the known distribution. For new domains (e.g., VinDr-PCXR), where X-ray embeddings deviated from the learned distribution (Figure 2), and X-rays do not align with known radiology reports of adults, results were more reliable when using *Template* ensembles without report information. Similarly, image ensembles did not improve the results, as the augmented views failed to align with the known distribution. To further assess the impact of our ensembling strategy, we additionally evaluated it on the FM *BiomedCLIP*. The results confirmed its effectiveness on adult datasets, while performance remained consistently low on pediatric cases in all ZS scenarios.

### 3.2. Comparing training-based methods to ZS ensemble method

In Figure 4, we compare the MCC of the best ZS ensemble method of *MedImageInsight* against our baselines (*DenseNet, CheXagent, RAD-DINO, Biomed-CLIP*) and all training-based methods across all three datasets. Appendix D.1, Figure 6 and Table 5 provide a

detailed comparison of all metrics for the baselines and training-based methods. Table 6 presents a quantitative analysis of the associated computational costs.

**MIMIC.** Among *MedImageInsight* methods, ZS ensembling performed best in low-data regimes (1% training data). However, *CheXagent*, trained on MIMIC, achieved the highest overall performance. BiomedCLIP and DenseNet consistently underperformed compared to ZS ensembling. Although *RAD-DINO* was pre-trained on MIMIC, it required at least 50% of training data to surpass our ZS approach.

**CheXpert.** Besides *CheXagent* (pre-trained on CheXpert), which is only outperformed by linear probing with 100% training data, ZS ensemble performed best when less than 10% of training data was available. With 50% data, linear probing exceeded ZS by 2% in MCC, while LoRA required 80% to surpass it by just 1%. *RAD-DINO*, despite being pre-trained on CheXpert, needed 100% data to outperform ZS ensemble. DenseNet, BiomedCLIP, and kNN performed worse than ZS ensemble regardless of training data availability.

**VinDr-PCXR.** While *CheXagent* and linear probing with 1% of training data, achieved the highest MCC, their TPR remained < 50%, indicating that it failed to reliably predict pneumonia. Hence, with only 1% training data, the ZS ensemble method still performed better than other methods. In general, LoRA fine-tuning achieved highest Acc with 86.6% using 50% data.

**Discussion.** For *MedImageInsight*, in low data regimes, the training-free ZS ensemble method led to best performance in all domains. It remained competitive even as training data increased, especially in adult datasets. With >10% annotated training data, linear probing improved performance, highlighting that *MedImageInsight* effectively captures clinically meaningful features. Notably, linear probing on *MedImageInsight* outperformed that of *RAD-DINO*, despite *RAD-DINO* being pre-trained on both MIMIC and CheXpert. On adult data, fine-tuning the image encoder caused catastrophic forgetting in low-data regimes and yielded only marginal improvements over ZS ensembling with more training data. In contrast, for pediatric cases, which are from an entirely new domain, fine-tuning with > 50% of training data captured the distribution shift from adults to pediatrics and outperformed other methods. The baselines *CheXagent* and *Biomed-CLIP* achieved TPR < 50% on VinDr-PCXR. Only the baseline DenseNet trained with > 80% of training data achieved a TPR and TNR > 50% and exceeded ZS ensemble performance in MCC. However, LoRA fine-tuning with 50% training data performed best overall on VinDr-PCXR. These findings highlight the robustness of *MedImageInsight* and the effectiveness of ZS ensemble method in low-data scenarios. If more than 10% (MIMIC, VinDR-PCXR) or 50% (CheXpert) of training data is available, along with sufficient computing resources, linear probing can provide slight improvements over ZS ensemble. However, in clinical settings, computational constraints often limit training-based adaptation. Only on VinDR-PCXR, LoRA fine-tuning remains the best option when training is feasible. Table 6 in Appendix D.1 compares the computational costs of ZS and training-based methods and DenseNet-121. ZS methods require no training time or labeled data, making them practical for resource-limited settings. Notably, ZS prompt ensembling is even faster than standard ZS, as embeddings are computed once from MIMIC and reused across experiments. As expected, training-based methods require labeled data and significantly more computational resources.

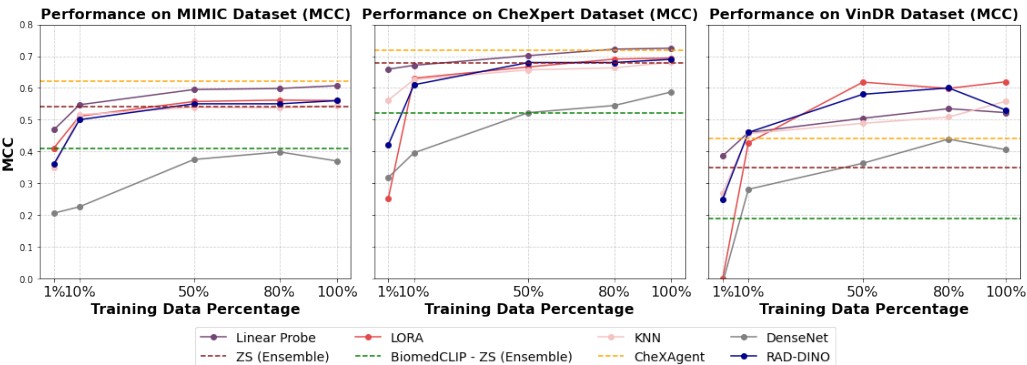

Figure 4: Comparison of most effective ZS (Ensemble) ((12) Template and Report (MIMIC, CheXpert); (10) Templates (VinDr-PCXR)) and training-based methods and baseline models across (left) MIMIC, (middle) CheXpert, (right) VinDr-PCXR using varying amounts of training data. ZS (Ensemble) shows competitive performance, especially in low-data regimes. Only baseline model, *CheXagent*, which was trained on all three datasets, consistently outperforms ZS (Ensemble).

### 3.3. Bias Assessment

Figure 3 (right) illustrates the absolute MCC differences across the demographic subgroups, sex, age, and race, in the CheXpert dataset. All evaluated methods exhibited varying levels of bias, reflected in performance differences across subgroups. ZS ensembling demonstrated a notable bias reduction compared to linear probing across all variables. For linear probing, differences ranged from 10% for sex, 14% for race, and up to 24% for age. In contrast, the ZS ensemble method achieved considerably lower bias levels, with the highest observed difference of 10% for race, while bias for sex and age remained below 5%. Although the ZS performance method demonstrated improved fairness compared to linear probing, methods such as LORA and KNN exhibited even smaller differences across all variables.

### 4. Conclusion

In this work, we evaluated strategies for an effective application of a state-of-the-art vision-language FM in clinical settings based on pneumonia prediction across diverse populations. By applying text and vision ensembles for ZS prediction, we achieved up to 43% improvement in MCC in adults and 35% in pediatrics compared to training-based methods in low-data scenarios. Also, compared to linear probing, ZS ensembling reduced biases related to sex, age, and race by 6%-17% (MCC). These findings demonstrate the potential of ZS ensembles as a resource-efficient alternative to training-based adaptation methods, especially in low-data and computationally constrained environments. In scenarios where data and computational resources are abundant, training-based methods such as linear probing or fine-tuning with LoRA are preferable for optimal performance. This work contributes to the equitable and accessible integration of FMs into clinical workflows, supporting diverse patient populations.

## Acknowledgments

LF is funded by the Global Glimpse Program of the University of Stuttgart, which is supported by the Deutsche Forschungsgemeinschaft (DFG) as part of the Excellence Strategy of the Federal and State Governments as well as by the Carl-Duisburg-Fellowship of the Bayer Foundation.

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

## Appendix A. Related Work

The applied FM, *MedImageInsight*, builds upon large-scale contrastive multimodal pre-training, which includes 14 different domains, including chest X-rays. Compared to other FMs such as (Zhang et al., 2023; Hyland et al., 2023; Bannur et al., 2024), *MedImageInsight* is trained on images, text, and labels, enabling adaptation to diverse distributions, such as adult and pediatric cases. While some Large Language Models (LLMs), like Med-Gemini (Saab et al., 2024) or Med-PaLM-M (Tu et al., 2024), are trained on text and labels, they are more than 10 times larger compared to *MedImageInsight*.

Foundation models trained in a contrastive manner generalize well on zero-shot (ZS) tasks by aligning image and text embeddings without task-specific training. However, the success of ZS performance depends on the quality of the text prompt. Radford et al. (Radford et al., 2021) highlighted the sensitivity of ZS performance to text prompts in general natural tasks and, therefore, introduced the idea of text prompt ensembles using up to 80 different templates and averaging them over the embedding space. They demonstrated improvements of almost 5% on the natural image dataset ImageNet (Deng et al., 2009).

To this end, Shu et al. (Shu et al., 2022) introduced Test-time Prompt Tuning (TPT), which generates image embedding ensembles on the fly based on multiple augmented versions of one input image. To exclude noisy augmentations, they added a confidence-based filter. Döbler et al. (Döbler et al., 2024) combined both approaches, creating text ensembles from templates and vision ensembles from augmented images, and tested them on general domains. However, their effectiveness in medical contexts, particularly for diverse patient populations, remains unexplored.

The application of FMs in clinical studies has revealed significant biases in their feature embeddings (Glocker et al., 2023; Santomartino et al., 2024). Glocker et al. (Glocker et al., 2023) found that FMs often encode demographic factors, which might lead to performance differences across subgroups. Mitigation strategies include adversarial training (Ganin et al., 2016), fairness-aware loss functions (Zafar et al., 2017), or the reduction of shortcut learning (Fay et al., 2023). In medical tasks, such biases are particularly concerning and need to be addressed to provide fair healthcare (Larrazabal et al., 2020).

Pneumonia detection from chest X-rays was studied for various model architectures in (Singh et al., 2024). They presented that different types of convolutional-based models, such as VGGs (Simonyan and Zisserman, 2014), ResNets (He et al., 2016), InceptionV3 (Szegedy et al., 2016), and DenseNets (Huang et al., 2017), perform worse than Vision Transformers (ViTs) (Alexey, 2020). However, in comparison, a Vision Transformer (ViT) has more than 85B trainable parameters, while the convolutional-based models have fewer than 200M. Likewise, the image encoder of *MedImageInsight* operates on 360M trainable parameters, offering a balance between efficiency and accuracy.

## Appendix B. Limitations

While in our study, we evaluated a variety of text prompts and ensemble strategies, it is possible that more effective templates or more complex prompt designs might further improve performance. Especially, the lack of pediatric-specific radiology reports likely constrained the performance on pediatric cases. Incorporating such domain-specific reports might yield better alignment and performance in pediatric populations. Additionally, while our study

demonstrates promising approaches for deploying embedding-based vision-language FMs in clinical settings, this study does not explore other FMs on a large-scale besides the three baseline FMs, *CheXagent*, *RAD-DINO*, and *Biomed-CLIP*. We limited our work to the prediction of pneumonia from chest X-rays, as this condition appears in both adult and pediatric patients. However, exploring further diseases as well as modalities could provide broader insights and reliability regarding the application of FMs in clinical settings and is a crucial next step of our work. Although our work explores the FM in three different environments, this may not fully reflect the heterogeneity of real-world clinical populations. In our upcoming work, we aim to address these limitations, especially by exploring more complex prompting strategies, as well as different domains and modalities, to enable a reliable and fair application of FMs in clinical workflows. Moreover, our future analysis will analyze misclassified cases, investigate failure patterns, and compare errors across different methods to provide further insights into model behavior and reliability.

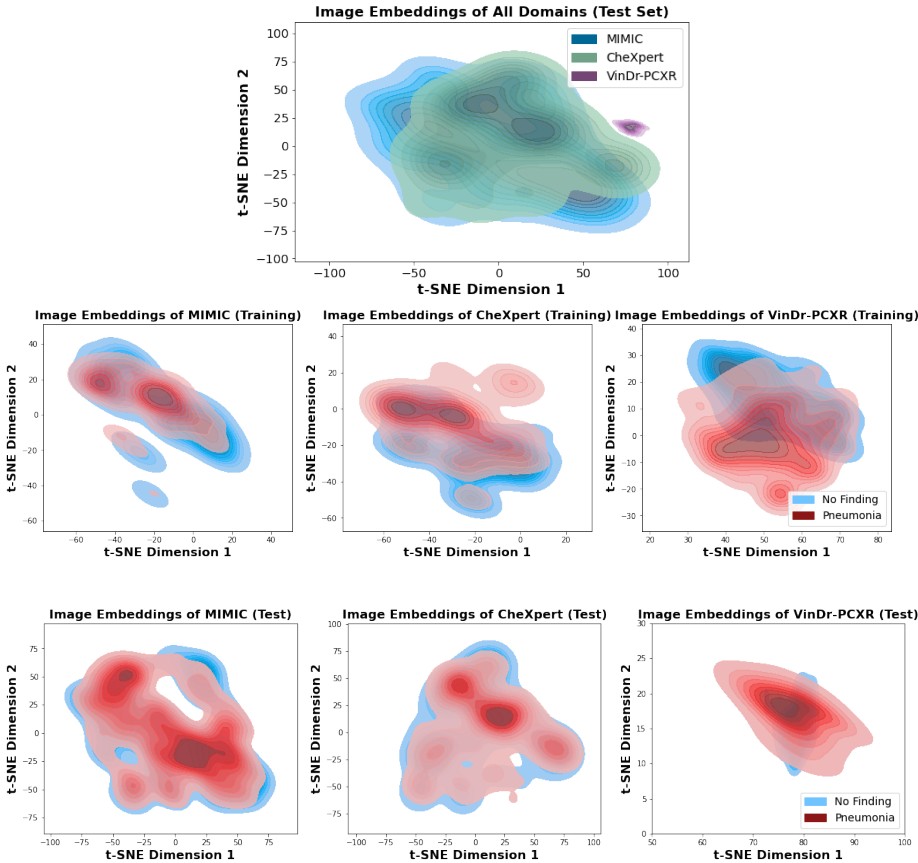

Figure 5: (Top) Comparison of X-ray embeddings of *MedImageInsight* across the test datasets of MIMIC (adults), CheXpert (adults), VinDr-PCXR (pediatrics) using t-SNE. (Middle) Comparison of the X-ray embeddings distribution of *No Finding* and *Pneumonia* across the training datasets of MIMIC (left), CheXpert (middle), VinDr-PCXR (right). (Bottom) Comparison for all test datasets.

Table 2: ZS performance of BiomedCLIP model for different text and vision prompts across adult (MIMIC, CheXpert), and pediatric (VinDr-PCXR) datasets (Acc, TNr, TPR in [%]

| | | Prompt | Vision Ens.[+] | MIMIC | | | | CheXpert | | | | VinDr-PCXR | | | |
|---|---|---|---|---|---|---|---|---|---|---|---|---|---|---|---|
| | | | | Acc | TNR | TPR | MCC | Acc | TNR | TPR | MCC | Acc | TNR | TPR | MCC |
| Template | 1 | | | 51.4 | 28.8 | 74.0 | 0.03 | 48.4 | 12.0 | 84.7 | -0.05 | 48.3 | 94.4 | 2.2 | -0.09 |
| | 2 | *chest* | | 38.0 | 29.1 | 47.0 | 0.24 | 32.1 | 17.0 | 47.2 | -0.38 | 38.2 | 67.4 | 9.0 | -0.29 |
| | 3 | *X-ray* | - | 43.0 | 44.4 | 41.9 | 0.14 | 37.6 | 33.2 | 42.0 | -0.24 | 49.4 | 94.4 | 4.5 | -0.03 |
| | 4 | *x. c.** | | 40.4 | 29.1 | 51.7 | 0.2 | 31.8 | 28.4 | 35.2 | -0.37 | 34.3 | 28.1 | 40.4 | -0.32 |
| | 5 | *x. c. a.** | | 40.2 | 19.4 | 60.7 | 0.22 | 31.2 | 22.9 | 39.5 | -0.38 | 45.5 | 3.4 | 87.7 | -0.17 |
| Report | 6 | Findings (F) | | 48.5 | 67.6 | 29.4 | 0.03 | 48.8 | 73.9 | 23.6 | 0.03 | 50.6 | 98.9 | 2.2 | 0.04 |
| | 7 | Impression (I) | | 59.5 | 94.5 | 24.5 | 0.26 | 61.4 | 91.8 | 31.0 | 0.29 | 53.4 | 94.4 | 12.4 | 0.12 |
| | 8 | F+I | - | 55.0 | 41.2 | 68.8 | 0.1 | 53.8 | 40.2 | 67.3 | 0.08 | 55.1 | 97.8 | 12.4 | 0.19 |
| | 9 | Full Report | | 52.7 | 60.3 | 45.1 | 0.05 | 47.9 | 70.5 | 25.4 | -0.05 | 39.3 | 48.3 | 30.3 | -0.22 |
| Prompt Ensemble | 10 | Templates | | 69.5 | 79.9 | 59.1 | 0.4 | 73.9 | 90.9 | 56.9 | 0.51 | 52.2 | 100 | 4.0 | 0.15 |
| | 11 | Reports | - | 61.7 | 64.6 | 58.8 | 0.23 | 69.0 | 82.7 | 55.3 | 0.4 | 53.4 | 100 | 6.7 | 0.19 |
| | 12 | Both | | 64.9 | 61.9 | 67.9 | 0.3 | 72.5 | 81.3 | 63.7 | 0.46 | 53.4 | 100 | 6.7 | 0.19 |
| | 10 | Templates | All | 79.1 | 80.7 | 59.6 | 0.41 | 74.6 | 92.1 | 57.1 | 0.52 | 52.2 | 100 | 4.4 | 0.15 |
| | 11 | Reports | All | 62.2 | 66.5 | 57.8 | 0.24 | 70.5 | 86.8 | 54.3 | 0.43 | 53.4 | 100 | 6.7 | 0.19 |
| | 12 | Both | All | 65.7 | 63.1 | 68.2 | 0.31 | 74.4 | 84.8 | 64.0 | 0.5 | 52.8 | 100 | 5.6 | 0.17 |
| | 10 | Templates | CF[++] | 70.0 | 80.6 | 59.4 | 0.41 | 74.6 | 92.1 | 57.1 | 0.52 | 52.2 | 100 | 4.5 | 0.15 |
| | 11 | Report | CF[++] | 62.3 | 66.7 | 57.8 | 0.25 | 70.4 | 86.4 | 54.4 | 0.43 | 52.8 | 100 | 5.6 | 0.17 |
| | 12 | Both | CF[++] | 65.7 | 63.3 | 68.1 | 0.31 | 74.3 | 85.0 | 63.6 | 0.5 | 52.8 | 100 | 5.6 | 0.17 |

*x.=X-ray, c.=chest, a.=anteroposterior; [+]Ens.= Ensemble; [++]CF=confidence-score filter

## Appendix C. Extended Dataset

In Figure 5 (top), we present the image embeddings of our balanced test datasets, in addition to those shown in Figure 2 (right). Furthermore, Figure 5 (middle/bottom) displays t-SNE plots of the training and test distributions, separated by the investigated classes *No Finding* and *Pneumonia*, across the adult datasets, MIMIC (left) and CheXpert (middle), as well as the pediatric dataset, VinDr-PCXR (right).

## Appendix D. Extended Results

### D.1. Evaluation Metrics

In our experiments, all test datasets are balanced to ensure fair evaluation of pneumonia prediction. We evaluate the performance focusing on accuracy (Acc), true-negative rate (TNR), true-positive rate (TPR), and Matthews correlation coefficient (MCC).

### D.2. Zero-shot Evaluation of FM *Biomed-CLIP*

To additionally evaluate the ZS ensembling strategy, we performed all ZS experiments on the embedding-based vision-language FM *BiomedCLIP* and provide the results in Table 2.

### D.3. Hyperparameter optimization

For our training-based methods k-NN and LoRA, we investigated a suitable number of k-neighbors as well as rank r for LoRA fine-tuning. We provide the results of our ablation study in Table 3 and Table 4, respectively. Based on this ablation study, we chose $k = 5$ and $r = 8$.

Table 3: Hyperparameter optimization: Comparison of KNN performance (accuracy in [%]) with different values of $k$ across MIMIC, CheXpert, and VinDr datasets. We chose $k = 5$ as it achieves best accuracy in most scenarios.

| Training Data | MIMIC | | | CheXpert | | | VinDr | | |
|---|---|---|---|---|---|---|---|---|---|
| | $k = 5$ | $k = 10$ | $k = 100$ | $k = 5$ | $k = 10$ | $k = 100$ | $k = 5$ | $k = 10$ | $k = 100$ |
| 0.01 | 67.30 | - | - | 76.65 | - | - | 56.75 | - | - |
| 0.1 | 75.70 | 75.05 | - | 81.10 | 80.25 | - | 70.25 | 67.40 | - |
| 0.5 | 76.95 | 77.50 | 77.80 | 82.60 | 82.30 | 82.00 | 73.60 | 71.35 | 67.40 |
| 0.8 | 76.85 | 78.30 | 77.90 | 83.00 | 82.70 | 82.15 | 74.70 | 75.20 | 69.65 |
| 1.0 | 77.20 | 78.75 | 77.80 | 83.90 | 83.30 | 82.60 | 76.95 | 75.25 | 70.25 |

Table 4: Hyperparameter optimization: Comparison of LoRA performance (accuracy in [%]) with different rank values $r$ across MIMIC, CheXpert, and VinDr datasets. We chose $r = 8$ as it achieves the best accuracy in most scenarios.

| Training data | MIMIC | | | CheXpert | | | VinDr | | |
|---|---|---|---|---|---|---|---|---|---|
| | $r = 8$ | $r = 16$ | $r = 32$ | $r = 8$ | $r = 16$ | $r = 32$ | $r = 8$ | $r = 16$ | $r = 32$ |
| 0.01 | 69.95 | 60.80 | 56.30 | 56.10 | 67.35 | 68.10 | 50.00 | 50.00 | 64.00 |
| 0.1 | 75.45 | 74.30 | 76.40 | 80.70 | 80.10 | 79.10 | 71.35 | 73.60 | 74.20 |
| 0.5 | 77.85 | 77.70 | 76.90 | 83.30 | 79.74 | 83.10 | 80.90 | 79.20 | 80.30 |
| 0.8 | 77.90 | 78.50 | 77.70 | 84.50 | 84.95 | 84.60 | 79.80 | 80.30 | 79.20 |
| 1.0 | 77.95 | 79.15 | 78.70 | 84.70 | 83.60 | 84.20 | 80.90 | 79.80 | 82.60 |

### D.4. Results Baselines and training-based methods.

In Figure 6 and Table 5, we provide additional results of the experiment shown in Section 3.2. The figure shows Acc, TNR, and TPR of our baselines *CheXagent*, *RAD-DINO*, and DenseNet-121 as well as the training-based methods explored on *MedImageInsight* using a varying amount of training data compared to our ZS (Ensemble) Method. We did not retrain CheXagent as it is already pre-trained on all three datasets.

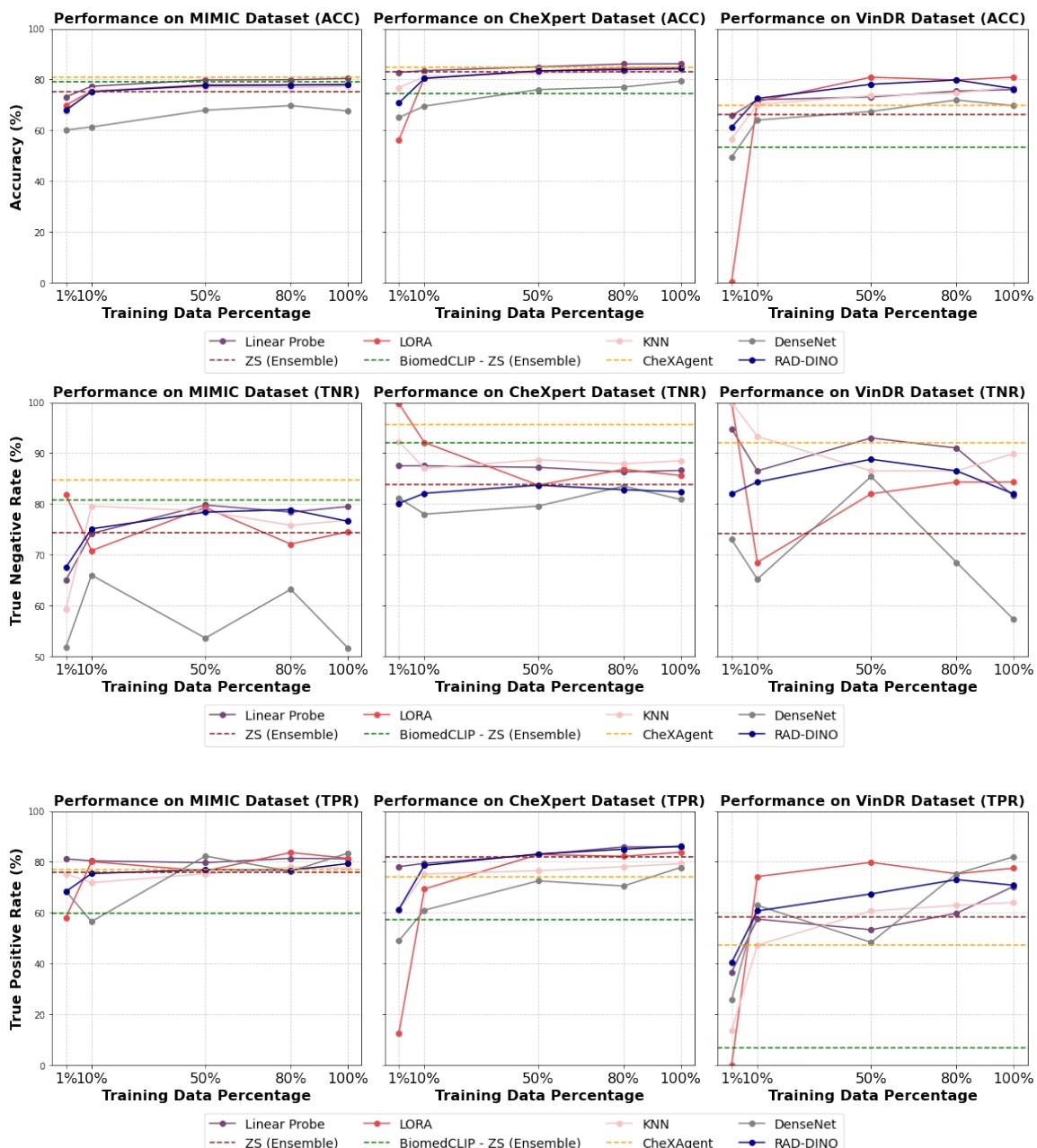

Figure 6: Comparison of Acc, TNR, TPR of ZS ensemble and baselines as well as training-based method across MIMIC, CheXpert, and VinDr-PCXR dataset.

## D.5. Computational Costs

Table 6 compares the computational costs of different MedImageInsight methods, highlighting the efficiency of zero-shot (ZS) approaches versus training-based methods. While ZS

Table 5: Comparison of Acc, TNR, TPR of ZS ensemble and baselines as well as training-based method across MIMIC, CheXpert, and VinDr-PCXR dataset.

| Method | Percentage | MIMIC | | | | CheXpert | | | | VinDr-PCXR | | | |
|---|---|---|---|---|---|---|---|---|---|---|---|---|---|
| | | Acc | TNR | TPR | MCC | Acc | TNR | TPR | MCC | Acc | TNR | TPR | MCC |
| *CheXagent* | | 80.8 | 84.7 | 77.0 | 0.62 | 84.9 | 95.7 | 74.1 | 0.72 | 69.7 | 92.1 | 47.2 | 0.44 |
| *RAD-DINO* | 0 | 50.2 | 51.3 | 49.1 | 0.0 | 50.3 | 48.7 | 52.0 | 0.0 | 56.7 | 86.5 | 27.0 | 0.17 |
| | 0.0 | 67.9 | 67.5 | 68.3 | 0.36 | 70.7 | 80.1 | 61.3 | 0.42 | 61.2 | 82.0 | 40.4 | 0.25 |
| | 0.1 | 75.2 | 75.1 | 75.4 | 0.50 | 80.4 | 82.1 | 78.6 | 0.61 | 72.5 | 84.3 | 60.7 | 0.46 |
| | 0.5 | 77.6 | 78.4 | 76.9 | 0.55 | 83.4 | 83.7 | 83.1 | 0.68 | 78.1 | 88.8 | 67.4 | 0.58 |
| | 0.8 | 77.8 | 78.9 | 76.7 | 0.55 | 83.9 | 82.8 | 85.0 | 0.68 | 79.8 | 86.5 | 73.0 | 0.6 |
| | 1.0 | 77.9 | 76.6 | 79.3 | 0.56 | 84.3 | 82.4 | 86.2 | 0.69 | 76.4 | 82.0 | 70.8 | 0.53 |
| *Linear Probe* | 0.01 | 73.1 | 65.0 | 81.2 | 0.47 | 82.8 | 87.5 | 78.1 | 0.66 | 65.7 | 94.8 | 36.6 | 0.39 |
| | 0.1 | 77.3 | 74.2 | 80.4 | 0.55 | 83.5 | 87.5 | 79.4 | 0.67 | 72.0 | 86.5 | 57.5 | 0.46 |
| | 0.5 | 79.8 | 79.8 | 79.7 | 0.60 | 85.0 | 87.2 | 82.9 | 0.70 | 73.1 | 93.0 | 53.3 | 0.50 |
| | 0.8 | 79.9 | 78.4 | 81.4 | 0.60 | 86.1 | 86.3 | 85.9 | 0.72 | 75.4 | 91.0 | 59.8 | 0.53 |
| | 1.0 | 80.4 | 79.5 | 81.2 | 0.61 | 86.2 | 86.6 | 85.9 | 0.73 | 76.0 | 81.6 | 70.3 | 0.52 |
| *LoRA* | 0.01 | 69.9 | 81.8 | 58.1 | 0.41 | 56.1 | 99.8 | 12.4 | 0.25 | 0.5 | 100.0 | 0.0 | 0.00 |
| | 0.1 | 75.4 | 70.8 | 80.1 | 0.51 | 80.7 | 92.1 | 69.3 | 0.63 | 71.3 | 68.5 | 74.2 | 0.43 |
| | 0.5 | 77.9 | 79.3 | 76.4 | 0.56 | 83.3 | 83.7 | 82.9 | 0.67 | 80.9 | 82.0 | 79.8 | 0.62 |
| | 0.8 | 77.9 | 72.1 | 83.7 | 0.56 | 84.5 | 86.8 | 82.2 | 0.69 | 79.8 | 84.3 | 75.3 | 0.60 |
| | 1.0 | 78.0 | 74.5 | 81.4 | 0.56 | 84.7 | 85.6 | 83.8 | 0.69 | 80.9 | 84.3 | 77.5 | 0.62 |
| *KNN* | 0.01 | 67.3 | 59.3 | 75.3 | 0.35 | 76.7 | 92.3 | 61.0 | 0.56 | 56.7 | 100.0 | 13.5 | 0.27 |
| | 0.1 | 75.7 | 79.6 | 71.8 | 0.52 | 81.1 | 87.0 | 75.2 | 0.63 | 70.2 | 93.3 | 47.2 | 0.46 |
| | 0.5 | 77.0 | 78.6 | 75.3 | 0.54 | 82.6 | 88.7 | 76.5 | 0.66 | 73.6 | 86.5 | 60.7 | 0.49 |
| | 0.8 | 76.9 | 75.8 | 77.9 | 0.54 | 83.0 | 87.9 | 78.1 | 0.66 | 74.7 | 86.5 | 62.9 | 0.51 |
| | 1.0 | 77.2 | 76.8 | 77.6 | 0.54 | 83.9 | 88.5 | 79.3 | 0.68 | 77.0 | 89.9 | 64.0 | 0.56 |
| *DenseNet-121* | 0.01 | 60.1 | 51.8 | 68.5 | 0.21 | 65.0 | 81.2 | 48.9 | 0.32 | 49.4 | 73.0 | 25.8 | -0.01 |
| | 0.1 | 61.3 | 66.0 | 56.5 | 0.23 | 69.5 | 78.0 | 61.0 | 0.40 | 64.0 | 65.2 | 62.9 | 0.28 |
| | 0.5 | 67.9 | 53.6 | 82.3 | 0.37 | 76.0 | 79.6 | 72.5 | 0.52 | 67.4 | 85.4 | 48.3 | 0.36 |
| | 0.8 | 69.7 | 63.2 | 76.3 | 0.40 | 77.0 | 83.5 | 70.5 | 0.54 | 71.9 | 68.5 | 75.3 | 0.44 |
| | 1.0 | 67.6 | 51.7 | 83.4 | 0.37 | 79.3 | 80.9 | 77.8 | 0.59 | 69.7 | 57.3 | 82.0 | 0.41 |

methods require no training, their inference costs vary, with ZS vision ensemble being the most computationally intensive. In contrast, training-based methods like linear probing and LoRA introduce training overhead.

Table 6: Computational cost comparison for training and inference of MedImageInsight's ZS and training-based methods using one Nvidia GeForce RTX 3090 - 24GB GPU.

| Model | ZS | ZS Prompt Ens.* | ZS Vision Ens.* | Linear Probe | LoRA | k-NN | DenseNet-121 |
|---|---|---|---|---|---|---|---|
| Train Parameters | 0 | 0 | 0s | 0.0021M | 360M | 0 | 7M |
| Training [per epoch] | 0s | 0s | 0 | 39.4s | 157.5s | 38.5s | 25.2s |
| Inference [per sample] | 0.06s | 0.056s | 35.3s | 0.056s | 0.056s | 0.06s | 0.01s |
| GFLOPs | 352.9 | 342.6 | 21926 | 342.6 | 342.6 | 342.6 | 26.6 |

*Ens.=Ensemble

