# OpenReview forum: "Beyond the Prompt: Deploying Medical Foundation Models on Diverse Chest X-ray Populations"
_MIDL.io/2025/Conference — MIDL 2025 Poster_

### Official Review · Reviewer_groK · 2025-02-16

**Confidence:** 4
**Preliminary Rating:** 2
**Final Rating:** 4

**Summary:**

The authors evaluate one foundation model called MedImageInsight for its performance in training-free ZS methods as well as training-based adaptation strategies. In this paper, they evaluate both in-domain and out-of-domain data.

**Strengths:**

(1) The authors use both in-domain datasets and out-of-domain dataset for testing the MedImageInsight model.

(2) The authors evaluate MedImageInsigt's performance with different amounts of training data.

**Weaknesses:**

This paper serves as an experimental extension to the original work: MedImageInsight: An Open-Source Embedding Model for General Domain Medical Imaging. The focus of this paper is on testing MedImageInsight with additional datasets. The original MedImageInsight paper is currently in pre-print status, with four pre-print citations to date.

Although the original MedImageInsight paper may be well-suited for MIDL, evaluating the original MedImageInsight paper falls outside the scope of this review. Assume the design of MedImageInsight is verified. Here are my comments:

(1) Lack of comparison with other state-of-the-art foundation models.

(2) Lack of deployment analysis for the model.

(3) The contributions are not clear.

**Detailed Comments:**

For point (1), only DenseNet-121 (developed in 2017 and 98% smaller than the image encoder of MedImageInsight) is used for comparison as a baseline. Please compare MedImageInsight with new state-of-the-art models, including foundation models.

For point(2), please include essential deployment information, such as inference speed analysis, deployment environment, and the potential of using the model for real-time application.

For point (3), most of the paper aims to verify the following points which are typical for foundation models:

(a) Testing on same-domain or similar-domain data, foundation models perform well; however, performance decreases on out-of-domain data.

(b) Foundation model performance improves with sufficient training data.

(c) Foundation models outperform baseline models with parameter and architectural design limitations.

Please highlight the contributions of the paper.

**Justification Of The Final Rating:**

I would love to thank the authors for their responses. With the newly added experiments, details for deployment analysis, and highlighted contributions, the quality of the paper improves significantly. I agree with the other reviewers that the paper can be further improved by a comparison of different training-based methods on OOD data.  I would love to change my rating from weak reject to weak accept.

One comment:

Please conduct a typo check for the final version of the paper.

One example: In Table 5, ACC can not be 167.9.

**Justification Of The Preliminary Rating:**

As the paper lacks comparisons with state-of-the-art methods and the contribution of the paper is not clear, I have to vote for weak reject. I encourage the authors to conduct major revisions to the paper.

**Questions To Address In The Rebuttal:**

This paper could potentially be classified as an application paper. Please address point (1) and point (3). Additionally, the use of the term 'deploy' in the paper may be misleading; therefore, addressing point (2) is also necessary.

**Special Issue:**

No

---

> ### Author Response · Authors · 2025-03-08
>
> We would like to thank Reviewer groK for providing valuable and insightful feedback on our manuscript. We address all concerns in the subsequent rebuttal and have revised the manuscript accordingly.
>
> > (1) Lack of comparison with other state-of-the-art foundation models.
>
> We acknowledge the need for additional comparisons with state-of-the-art foundation models. Our study is based on MedImageInsight, which has been publicly available since October 2024. Its lean architecture (<0.61B parameters) and promising results motivated us to use this model in our analysis. As suggested by the reviewer, we have now strengthened our analysis by extending our experiments to include three additional SOTA models with different architectures (Table 2, Table 5):
> * CheXagent (2024) (8B parameters): A vision-language model trained on 32 chest X-ray datasets. It generates free-text responses rather than embeddings, making our ensemble approach inapplicable. It is important to mention that all our datasets (CheXpert, MIMIC-CXR, VinDR-PCXR) have been part of the foundation model’s training. While it slightly outperforms MedImageInsight in overall accuracy (MIMIC: +3%, CheXpert: +1.2%), our ZS-Ensemble strategy ensures better TPR-TNR balance, particularly on CheXpert (21.4% vs. 10.7%). The TPR of VinDR-CXR is at 47.2%, hence, CheXagent performs worse compared to our ZS Ensembling approach with MedImageInsight.
> * RAD-DINO (2024) (86.6M parameters): A biomedical image encoder requiring a trained linear head. All our adult datasets (CheXpert, MIMIC-CXR) have been part of its training. To perform predictions a linear head (Methodology: Linear Probe) needs to be trained. It needs at least 50% of training data to surpass MedImageInsight with ZS-Ensembling.
> * BiomedCLIP (2023) (0.09B parameters): A CLIP-based vision-language model. While it could not outperform MedImageInsight, we could demonstrate that our ZS-Ensembling approach generalizes on another FM as it consistently outperforms other zero-shot methods on MIMIC and CheXpert.
>
> Please find all results in Section 3.1 and 3.2 as well as in the Appendix D.
>
> > (2) Deployment Analysis
>
> We appreciate the reviewer’s suggestion to clarify deployment aspects. In response, we have included an in-depth analysis of inference speed, resource requirements, and the feasibility of real-time applications in Table 6.
> * Deployment environment: MedImageInsight is open-source and runs locally or on Microsoft Azure. The model is built on PyTorch. The input image can be of any size as the framework inherits an automated image preprocessing. Regarding our ZS-Ensemble method, text embeddings are precomputed once (from MIMIC) and reused for all other OOD datasets, reducing overhead.
> * Real-Time Potential: The model achieves fast inference times (Table 6), with ZS methods requiring no training. Only vision ensembling introduces additional compute. Given these efficiency characteristics, MedImageInsight is suitable for real-time applications under computational constraints.
>
> > (3) Please highlight the contributions of the paper.
>
> We acknowledge the need to better highlight our contributions and have adapted them in our manuscript accordingly. While our study aligns with broader trends in FM adaptation, it goes beyond general observations by providing practical strategies for deploying and adapting FMs in real-world clinical settings, even when labeled data, time, or computational resources are limited. Specifically, we contribute:
>
> * ZS-Ensemble for Practical FM Deployment:
> Our proposed ZS ensembling method enables zero-shot FM deployment without additional training, making it feasible for settings with no labeled data, limited time, or constrained computational resources. By precomputing text ensemble embeddings (from MIMIC) and reusing them across datasets, our method enhances OOD robustness while keeping computational costs low. Our computational cost analysis (Table 6) offers practical insights for resource-constrained FM adaptation.
>
> * Adaptation Trade-offs and Bias Assessment:
> We provide a systematic comparison of adaptation strategies, showing that LoRA fine-tuning can degrade performance with insufficient data, while Linear Probing, despite its strong performance, introduces biases. Additionally, our bias assessment reveals fairness implications across demographic subgroups.
>
>
> * Multi-Site Evaluation and Generalization:
> We evaluate MedImageInsight across MIMIC-CXR, CheXpert, and VinDr-PCXR. By including pediatric datasets (VinDr-PCXR) in our study, we highlight the challenges of FM generalization across different age groups. Using ZS ensembles, we demonstrate a training-free approach to improve performance in both in-domain and out-of-domain distributions. When computational resources are available, Linear Probing enhances performance on adult datasets (closer to the training distribution), whereas fine-tuning is more effective for pediatric cases (out-of-distribution).

---

> > ### Comment · Area_Chair_6MYG · 2025-03-11
> >
> > Dear reviewer,
> >
> > The authors now wrote a rebuttal addressing your questions, could you please see if this satisfies your concerns and if needed adjust your score?

---

> > > ### Comment · Reviewer_groK · 2025-03-12
> > > **I have updated my score and my comments**
> > >
> > > The inclusion of newly added experiments, detailed deployment analysis, and highlighted contributions has significantly enhanced the quality of the paper. I would love to change my rating from weak reject to weak accept.
> > >
> > > I kindly suggest that the authors conduct a thorough typo check before submitting the camera-ready version of the paper.

---

### Official Review · Reviewer_MDU3 · 2025-02-16

**Confidence:** 4
**Preliminary Rating:** 3
**Final Rating:** 4

**Summary:**

This paper investigates techniques for prompting or fine-tuning of a medical foundation model (*MedImageInsight*) for pneumonia detection on chest X-ray images. The authors evaluate the performance of the model on three different datasets (one in-training data, one in-distribution, one out-of-distribution). They compare different zero shot techniques (different prompts, test-time augmentation of vision input, etc.) with training-based strategies (linear probing, LoRA, etc.). Through extensive experiments the authors find that on some data ZS can be improved by using prompt ensembles. On OOD pediatric data, linear probing and LoRA outperform ZS ensembles with sufficient training data.

**Strengths:**

Overall, the paper is well written and quite easy to follow. The topic is of very high practical relevance to the community. I also want to appreciate that the authors conducted a large amount of experiments.

**Weaknesses:**

**Limited scope:**

The paper is in an interesting position. On the one hand it tackles a very general question: "What is a good practice for deploying Medical Foundation Models on diverse, potentially OOD data?". This question is extremely relevant to the community and stakeholders. But then, authors narrow down the scope to a single foundation model (*MedImageInsight*), single domain (Chest X-ray) and single task (pneumonia detection). I understand that already this gives rise to a lot of different setups (different prompt / training strategies / datasets). But I believe that in the end, a practical solution to deploying foundation models must at least generalize to multiple domains and tasks, given that broad applicability is one of the main benefits of foundation models in the first place. But to me, the generalizability of the authors' findings is unclear, limiting the ability of the paper to answer above question. The authors find that their proposed ensemble method (prompt ensembles + vision ensemble) does not even provide consistent improvements over different populations, within the same modality, anatomical region, and task.

**Conclusion not supported by experiments:**

Related to the previous point, I find that most results are not very conclusive.
- In the abstract the authors write:
    > This ensembling strategy also promotes resource-efficient equitable clinical use by supporting diverse demographic subgroups, achieving MCC improvements of 6% by sex, 17% by age, and 10% by race compared to Linear Probe.

    This is true, but also Linear Probing performed worryingly bad on the fairness experiment and apart from one other prompt/training strategy - subgroup combination (ZS (Impression) - Age), all other techniques outperformed the proposed ZS (Ensemble) strategy in this regard (in particular LoRA / kNN performed much better). This is further corroborated by the fact that on the pediatric data (basically an extreme case of the age subgroup) fine-tuning performance was quite close to performance on adult data, whereas ZS (Ensemble) showed a significant drop in performance.

- The bad performance of ZS (Ensemble) on pediatric data is discussed in the paper:
    > For new domains (e.g., VinDr-PCXR), where X-ray embeddings deviated from the learned distribution (Figure 2), and X-rays do not align with known radiology reports of adults, results were more reliable when using Template ensembles without report information. Similarly, image ensembles did not improve the results, as the augmented views failed to align with the known distribution.

    But actually, if I am not mistaken, Table 1 even tells us that on pediatric data the simple template prompt (2) outperforms all ensemble strategies (including template ensemble).

- Overall, authors seem to really believe in the ZS (Ensemble) strategy. In the discussion they write:
    > By applying text and vision ensembles for ZS prediction, we achieved up to 43% improvement in MCC in adults and 35% in pediatrics compared to training-based methods in low-data scenarios. Also, we reduced biases related to sex, age, and race by 6%-17% (MCC) compared to Linear Probe.

I come to a different conclusion after reading the paper and looking at the experimental results. I would say that (assuming enough training data) finetuning using LoRA seems to be the most promising strategy, given that it performs much more consistently across different populations and different subgroups. In the fairness experiments, while ZS (Ensemble) performs better than linear probing, it does not outperform non-ensemble ZS (Impression) and seems to perform much worse than LoRA or KNN.

**Detailed Comments:**

- Please make sure that all appendix sections are referenced in the main paper. In particular, it took me some time to figure out that the authors actually discuss limitations of their work in the appendix. This should clearly be referenced in the main paper (ideally the main ones are also discussed in the main paper at least to some extent).
- Following sentence in Sec. 3.1: "Most other templates resulted even in either TPR or TNR below 50%" is a bit confusing. I think the authors mean that most other templates resulted in either TPR or TNR below 50%.
- Figure 3 right plot:
    - Colors are very hard to distinguish
    - How were the shown values aggregated, e.g. for subgroup race, there are 3 different categories.
    - Why did the authors choose to show these specific methods and not e.g. ZS (Template), ZS (Report)?
    - Caption: I think that y-axis is showing absolute difference of mean MCCI (not "absolute mean MCC across subgroup")
- What does the softmax in Figure 1 mean? Why would you feed the cosine similarity through a softmax? I also do not find where this is mentioned in the text.
- How much training data is used for training-based approaches? 744 samples? How exactly did the authors balance the training data for each of the datasets?
- For comparison with training-based methods, the authors write:
    > In Figure 4, we compare the MCC of the best ZS ensemble method against various training-based methods across all three datasets

    Did they use the same ZS ensemble method for all datasets, even though on pediatric data template ensemble performed better than (template ensemble + vision ensemble)? Also, did they use vision ensemble (All) or vision ensemble (CF), as these perform very similar on MIMIC / CheXpert data?
- I assume that at test time, the proposed ensemble method require more computation than non-ensemble prompting strategies or training-based adaption methods. I think this might be a very important point to consider in practice. Can the authors comment on this?
- Unfortunately, *MedImageInsight* is not open-source and can only be used via API calls to Microsoft Azure, for which a subscription is needed. This means that results in this paper are not independently reproducible. I understand that this is probably not the authors' fault but would appreciate if this limitation would be openly acknowledged in the manuscript.

**Justification Of The Final Rating:**

The authors addressed most of my concerns in their revision. In particular, the additional experiments using a second FM, BiomedCLIP, are greatly appreciated. However I am still not convinced that the proposed ZS ensembling strategy is a good alternative to a training-based approach *if* additional training data is available. To me this is also an interesting experimental result, that is worth being published (together with the experimental comparisons with other ZS strategies).

**Justification Of The Preliminary Rating:**

**Why not lower?**

The topic is of extremely high relevance and the authors performed extensive and well-designed experiments. The paper is well written and easy to follow.

**Why not higher?**

I do not think that the authors' conclusion is supported by the experiments. I am also skeptical that the findings would generalize to other domains and tasks.

**Questions To Address In The Rebuttal:**

I would appreciate if the authors could further elaborate on their conclusion. Maybe I missed something in the experiments section that would support their claim that ZS (Ensemble) is the best strategy for deploying foundation models on diverse populations.

---

> ### Author Response · Authors · 2025-03-08
>
> We would like to thank Reviewer MDU3 for its detailed and valuable feedback. We address all concerns in the subsequent rebuttal and have revised the manuscript accordingly.
>
> > [...] a practical solution to deploying foundation models must at least generalize to multiple domains and tasks [...]
>
> While our study focuses on a single FM (MedImageInsight), domain (chest X-ray), and task (pneumonia), this choice is guided by practical and scientific considerations. Pneumonia is clinically significant and present in both adult and pediatric datasets, allowing us to assess FM generalization across different populations. To further demonstrate the broad applicability of our approach, we expanded our analysis to include Biomed-CLIP (Table 2), confirming that ZS ensembling consistently achieves the best ZS performance on adult datasets.  The performance on pediatric cases using Biomed-CLIP is in no zero-shot scenario valid. In future work, we extend our work to additional domains and tasks.
>
> > [...] their proposed ensemble method [...] does not even provide consistent improvements [...]
>
> We acknowledge performance variations across datasets. However, ZS ensembling achieved the best ZS performance across all datasets. As noted in our manuscript, MIMIC and CheXpert embeddings are more similar, allowing precomputed report ensembles (from MIMIC) to perform well on adult datasets. In contrast, pediatric data differs significantly from adult distributions, making template-based reports more effective.
>
> Our findings contribute to understanding FM deployment limitations and ensembling trade-offs, offering insights relevant to real-world clinical settings.
>
> > Conclusion no supported by experiments: Bias assessment
>
> * Our bias analysis on CheXpert highlights a key trade-off: while Linear Probing is computationally efficient and outperforms ZS-Ensemble, it introduces biases. While fine-tuning can mitigate these biases, it requires significantly more training data and computational resources, making it less practical for real-world deployment (Table 6). We acknowledge that LoRA and kNN perform well given sufficient data, but they require at least 50%–100% of the training data to surpass ZS performance.
>
> * Regarding pediatric performance (Table 1), while the simple template prompt achieves higher accuracy/MCC, its TPR is close to random (52.8%, 41.6%), whereas ensemble prompts lead to a more balanced TPR/TNR above 60%. This indicates that ensembles promote more stable predictions rather than overfitting to dominant class distributions.
>
> * We agree that, given sufficient labeled training data, LoRA fine-tuning or linear probing may be preferable. However, in case of LoRA, at least 50\% of training data for adult datasets have to be available to perform better than ZS Ensembling. In cases where linear probing is used, bias investigations need to be taken into account.
> A major contribution of our work includes to explore how FM can be applied in clinical settings with minimal adaptation. A key advantage of ZS Ensembling is that no adaptation is required. The text ensemble only needs to be created once, which can even happen outside of clinical practice, as the ensembles are based on templates and MIMIC reports in all scenarios.
>
> > Please make sure that all appendix sections are referenced in the main paper.
>
> We adapt our manuscript and explicitly reference limitations in thee introduction. Due to space constraints, we did not include them in the main paper.
>
> > Figure 3
>
> We adapted colors and y-axis. Aggregation was performed by computing differences between all groups (Asian-White, Asian-Black, Black-White) and reported mean difference. Fig. 3 is based on CheXpert, we chose best ZS methods (without ensembling), which was (7) Impression for CheXpert.
>
> > Softmax
>
> We agree, softmax is not needed in the case of our binary prediction. We removed it as it is not necessary in the scope of this work.
>
> > Training data
>
> To ensure comparable training conditions, we matched the pneumonia case count to the smallest dataset. Each dataset included 342 pneumonia and 342 non-pneumonia cases (744 samples total). MIMIC and CheXpert cases were randomly sampled, while all VinDR pneumonia cases were used with random non-pneumonia sampling. Experiments were repeated five times.
>
> > ZS ensemble methods in Figure 4
>
> We used the best performing ZS ensemble method for each dataset. In case of similar performance, we applied vision ensemble (All) in the manuscript. However, results were similar for both.
>
> > Computational time
>
> For prompt ensembling, no additional test-time cost occurs since embeddings are precomputed once from Templates and MIMIC reports and reused across datasets. Vision ensembling increases computational time by generating 63 augmented images per input. Table 6 clarifies training, inference time, and computational costs.
>
> > Open-source
>
> MedImageInsight is fully open-source. It can be accessed via Microsoft Azure or run locally (as we did).

---

> > ### Comment · Reviewer_MDU3 · 2025-03-10
> >
> > I thank the authors for their detailed rebuttal and particularly for conducting additional experiments using a second FM, BiomedCLIP.
> >
> > **Robustness of ZS ensembling**
> >
> > Also on BiomedCLIP, we see that ZS ensembling (as all other prompting strategies) is not robust to OOD data. Here, the comparison to training-based methods is missing, but I assume it might be similar to that of MedImageInsight. Can the authors explicitly acknowledge in the manuscript that given sufficient training data, training-based methods are preferable over ZS-based methods in terms of OOD robustness.
> >
> > **Open-source code**
> >
> > > MedImageInsight is fully open-source. It can be accessed via Microsoft Azure or run locally (as we did).
> >
> > Thanks for providing a link to your code in the updated manuscript. Can you also point me to where users can download the network weights `medimageinsigt-v1.0.0.pt` and `language_model.pth`, both of which are necessary to run your code?

---

> > > ### Author Response · Authors · 2025-03-11
> > >
> > > We thank the reviewer for the feedback.
> > >
> > > > Robustness of ZS ensembling
> > >
> > > Due to time constraints, we have not explicitly compared BiomedCLIP’s ZS ensembling to training-based methods in the OOD setting. However, we anticipate that the trends observed with MedImageInsight would likely hold. Following your suggestion, we will add the following sentence to our conclusion:
> > > *In scenarios where data and computational resources are abundant, training-based methods such as linear probing or fine-tuning with LoRA are preferable for optimal performance.*
> > >
> > > > Open-source code
> > >
> > > All necessary data can be either downloaded from Hugging Face (https://huggingface.co/lion-ai/MedImageInsights) or directly from Microsoft Azure (https://ml.azure.com/registries/azureml/models/MedImageInsight/version/6?tid=e1475833-dbe8-4ae0-894e-cd0f561c98f4#artifacts).
> > >
> > > We have added both links to the README in our GitHub repository.

---

### Official Review · Reviewer_rWu7 · 2025-02-20

**Confidence:** 4
**Preliminary Rating:** 4
**Recommendation:** Poster
**Final Rating:** 4

**Summary:**

The authors use MedImageInsight, an open-source lightweight medical foundation model, to compare the performances of multiple zero-shot and training-based strategies. The experiments are conducted for pneumonia classification with three publicly available datasets including adult and pediatric patients. The techniques are evaluated on multiple metrics and aspects such as fairness and performance under low training data settings.

**Strengths:**

- The authors perform a thorough comparison of different approaches on multiple datasets and with several metrics.

- The approaches are also evaluated on potential sex, age and gender biases which is an important consideration.

- The different techniques are clearly explained.

**Weaknesses:**

- I believe the training-based methods were trained with a single training split per dataset and training data percentage. It could be insightful to also use multiple splits to evaluate the stability of the methods when changing the training data, especially in low data percentage settings.

- As the severe consequences of incorrect diagnoses are mentioned in the introduction, it could be nice to get some analysis of the errors to potentially evaluate their severity. Also if the errors are the same or different across the approaches.

- For reproducibility consideration, it would be nice to make the code publicly available.

**Detailed Comments:**

- Appendix A. related works and Appendix B. limitations should not be part of the appendix but part of the main text.

- The colour scheme in Figure 3 (right) could be different for a better contrast

**Justification Of The Final Rating:**

The authors perform a nice analysis of the different strategies for foundation models over several important aspects. I appreciated the clarification on the number of runs and on the variability. The authors also made the code publicly available. I believe more analysis of the errors would be insightful to potentially reveal more differences or similarities between the different methods.

**Justification Of The Preliminary Rating:**

The authors investigate an important question for vision-language foundation models. The paper is well-written and clear. The analysis covers several important aspects but others could have been investigated for a better comparison of the different approaches.

**Questions To Address In The Rebuttal:**

- How did you split/select the samples for the complete training sets and when reducing the training data percentage?
- Besides the performance metrics, did you perform some analysis of the errors made by the different methods?

---

> ### Author Response · Authors · 2025-03-07
>
> We would like to thank Reviewer rWu7 for providing valuable and insightful feedback on our manuscript. We address all concerns in the subsequent rebuttal and have revised the manuscript accordingly.
>
> >How did you split/select the samples for the complete training sets and when reducing the training data percentage?
>
> Thank you for this comment. Rather than using a single split, we performed random sampling from the training data in low-data regimes while ensuring class balance. Each experiment was run five times and we reported the mean value. We have now clarified this in the manuscript.
>
> > Besides the performance metrics, did you perform some analysis of the errors made by the different methods?
>
> We did not perform a detailed error analysis across different methods in this study. Our primary focus was on evaluating overall performance and computational efficiency. However, we agree that a deeper investigation into the types of errors made by each method could provide valuable insights and will explore this in future work.
>
> > For reproducibility consideration, it would be nice to make the code publicly available.
>
> Thanks for mentioning code publication. Our code is now publicly available under: https://github.com/loufay/Beyond-the-prompt. We also added the link to the manuscript.
>
> > Appendix A. related works and Appendix B. limitations should not be part of the appendix but part of the main text.
>
> We have now referenced the related work and limitations at the end of the introduction. However, due to space constraints, we were unable to include both in the main paper.
>
> > The colour scheme in Figure 3 (right) could be different for a better contrast
>
> Thank you for pointing this out. We adjusted the color scheme of Figure 3 (right) to improve contrast and readability.

---

> > ### Comment · Reviewer_rWu7 · 2025-03-10
> >
> > Thanks for the responses to the review comments and for publishing the code.
> >
> > Regarding the data split and the number of runs. As you have multiple runs, it may be relevant to add the variability in Figure 4 and Figure 6. Especially with low percentages of training data, I'm curious to know whether the performance of the training-based methods can show a lot of variability and can sometimes outperform ZS methods or have much worse results based on the randomly selected data.

---

> > > ### Author Response · Authors · 2025-03-11
> > >
> > > Thank you for providing us feedback to our comments.
> > >
> > > > Regarding the data split and number of runs. [...]
> > >
> > > We have included standard deviation in our updated plots for Figures 4 and 6 for the ZS (Ensemble) and the training-based methods (Linear Probe, LoRA fine-tuning, k-NN). The updated plots are available in our GitHub repository: https://github.com/loufay/Beyond-the-prompt/blob/main/Results/Results.md
> > >
> > > As noted by the reviewer, variability tends to be higher at lower training percentages. However, the overall trend remains the same: training-based methods do not outperform ZS (Ensemble) in the low-data scenario, nor do they perform substantially worse (as reflected by the mean values).

---

### Official Review · Reviewer_nDfA · 2025-02-22

**Confidence:** 3
**Preliminary Rating:** 4
**Recommendation:** Oral
**Final Rating:** 4

**Summary:**

This paper explores the deployment of foundation models in clinical settings with various approaches. The authors evaluate MedImageInsight for predicting pneumonia from chest X-rays with different prompt engineering and adaptation methods. Key findings include the significant impact of prompt design on ZS performance and the effectiveness of text and vision embedding ensembles, which outperform training-based methods in low-data scenarios.

**Strengths:**

1. The paper is easy to follow and well-organized.
2. It provides a comprehensive evaluation of prompt design and ensemble strategies for ZS prediction. The experiments are well-structured with appropriate baseline comparisons.

**Weaknesses:**

It would be good to discuss methodological choices (e.g. hyperparameter choices) through ablation studies for empirical justification. Moreover, the evaluation is limited to only pneumonia detection. Testing on additional pathologies would better show generalizability.

**Detailed Comments:**

The idea is clearly presented, and the paper is fairly clear in its approach and results.

**Justification Of The Final Rating:**

The addition of the ablation result clarifies the methodological decisions. While evaluation remains limited to pneumonia detection, your justification is reasonable, and the planned extensions would strengthen future work. I would like to maintain the rating of weak accept.

**Justification Of The Preliminary Rating:**

This paper is innovative for deploying foundation models in clinical settings through zero-shot ensemble strategies, achieving significant performance gains and bias reduction. However, it would be good to have a deeper discussion on the chosen techniques and the performances.

**Questions To Address In The Rebuttal:**

Can the proposed ensemble strategy be applied to other medical FMs or tasks beyond pneumonia prediction?

**Special Issue:**

No

---

> ### Author Response · Authors · 2025-03-07
>
> We would like to thank Reviewer nDfA for providing helpful feedback to improve our manuscript. We address all concerns in the subsequent rebuttal and have revised the manuscript accordingly.
>
> > It would be good to discuss methodological choices (e.g. hyperparameter choices) through ablation studies for empirical justification.
>
> We appreciate the reviewer’s suggestion to include ablation studies for hyperparameter choices. In our experiments, we empirically optimized the hyperparameters for the number of neighbors of kNN (k∈{5,10,100}) and the LoRA rank (r∈{8,16,32}). We selected k=5 and r=8, respectively, as they achieved the best performance across most experiments. We added these ablation results in Appendix D, Table 3, 4. Additionally, due to computational constraints, we selected the largest batch size that fit within our available resources (batch size = 8) for LoRA fine-tuning. We agree that further ablation studies could provide additional insights and we plan to explore this in future work.
>
> > Can the proposed ensemble strategy be applied to other medical FMs or tasks beyond pneumonia prediction?
>
> Thank you for raising this concern.
> * Our approach is applicable to other embedding-based vision-language models. To demonstrate this, we expanded our analysis to include Biomed-CLIP (Appendix D.2, Table 2), confirming that zero-shot ensembling consistently achieves the best performance for adults among the zero-shot results. Performance on pediatric datasets using Biomed-CLIP was low in all cases.
>
> * We focused on pneumonia detection in this study due to its clinical relevance, availability of well-curated datasets in both adult and pediatric cohorts. We acknowledge the importance of broader validation and plan to extend our evaluation to additional pathologies in future work to further assess the generalizability of our approach across different medical conditions.

---

> > ### Comment · Area_Chair_6MYG · 2025-03-11
> >
> > Dear reviewer,
> >
> > The authors now wrote a rebuttal addressing your questions, could you please see if this satisfies your concerns and if needed adjust your score?

---

> > ### Comment · Reviewer_nDfA · 2025-03-14
> >
> > The addition of the ablation result clarifies the methodological decisions. While evaluation remains limited to pneumonia detection, your justification is reasonable, and the planned extensions would strengthen future work. I would like to maintain the rating of weak accept.

---

### Author Rebuttal · Authors · 2025-03-07

**Rebuttal:**

Zip folder contains:

* Updated manuscript
* Updated manuscript with changes highlighted
* Latex project including all material

**Supporting Material:**

/attachment/dbd9b96694def5035f51884f830228e37512eb5a.zip

---

### Meta-Review · Area_Chair_6MYG · 2025-03-18

**Recommendation:** Accept (Poster)
**Confidence:** 4

**Metareview:**

The reviewers are overall in agreement about the merits of the paper, in particular the relevance of the topic and clarity of organization. Multiple reviewers pointed out the limited generalizability of the results (pneumonia data, evaluation setup), however the authors addressed most of the concerns in the rebuttal.

To add my own comment on this as the AC, I believe pneumonia is a differential diagnosis and therefore requires more than just imaging data (previous studies show low inter-observer agreement of radiologists using only imaging). In that sense it makes it MORE relevant for vision-language models, I feel that these points could be emphasized in 1-2 sentences in the paper.

The reviewers also suggest some further experiments which I believe may be helpful to the authors in future work.

Based on the overall reviews, scores and rebuttal process I recommend accepting the paper.